# Mesenchymal Stromal Cell-Derived Extracellular Vesicles for Vasculopathies and Angiogenesis: Therapeutic Applications and Optimization

**DOI:** 10.3390/biom13071109

**Published:** 2023-07-12

**Authors:** Ying Zhu, Zhao-Fu Liao, Miao-Hua Mo, Xing-Dong Xiong

**Affiliations:** 1Guangdong Provincial Key Laboratory of Medical Molecular Diagnostics, The First Dongguan Affiliated Hospital, Guangdong Medical University, Dongguan 523808, China; 2School of Medical Technology, Guangdong Medical University, Dongguan 523808, China

**Keywords:** mesenchymal stromal cells, extracellular vesicles, vasculopathies, angiogenesis, optimization

## Abstract

Extracellular vesicles (EVs), as part of the cellular secretome, have emerged as essential cell–cell communication regulators in multiple physiological and pathological processes. Previous studies have widely reported that mesenchymal stromal cell-derived EVs (MSC-EVs) have potential therapeutic applications in ischemic diseases or regenerative medicine by accelerating angiogenesis. MSC-EVs also exert beneficial effects on other vasculopathies, including atherosclerosis, aneurysm, vascular restenosis, vascular calcification, vascular leakage, pulmonary hypertension, and diabetic retinopathy. Consequently, the potential of MSC-EVs in regulating vascular homeostasis is attracting increasing interest. In addition to native or naked MSC-EVs, modified MSC-EVs and appropriate biomaterials for delivering MSC-EVs can be introduced to this area to further promote their therapeutic applications. Herein, we outline the functional roles of MSC-EVs in different vasculopathies and angiogenesis to elucidate how MSC-EVs contribute to maintaining vascular system homeostasis. We also discuss the current strategies to optimize their therapeutic effects, which depend on the superior bioactivity, high yield, efficient delivery, and controlled release of MSC-EVs to the desired regions, as well as the challenges that need to be overcome to allow their broad clinical translation.

## 1. Introduction

The vasculature is a well-organized network circulating the blood throughout the body and plays an important role in maintaining organ homeostasis by mediating the exchange of oxygen, nutrients, and waste within and between tissues. Abnormal conditions of the blood vessels can often lead to severe organ disabilities and even death. Therefore, new effective methods for preventing and treating vascular disorders are needed.

Mesenchymal stromal cells (MSCs) have recently appeared as novel therapeutics for a wide range of diseases. MSCs are fibroblast-like cells residing in diversetissues, such as bone marrow, adipose tissue, umbilical cord, and dental pulp, that have been indicated to have a perivascular origin [1]. According to the minimal defining criteria of the International Society for Cellular Therapy, cultured MSCs must be plastic adherent; express CD105, CD73, and CD90 but not CD45, CD34, CD14 or CD11b, CD79α or CD19, and human leukocyte antigen-DR surface molecules; and possess tri-lineage differentiation potential [2]. In addition, tissue-resident MSCs are considered CD34+ stromal cells/fibroblasts/fibrocytes/telocytes and lose CD34 expression in successive culture passages [3,4]. Resident/native CD34+ stromal cells/progenitor cells lose the expression of CD34 and are a source of alpha+ SMA myofibroblasts during different stages of repair [5]. MSCs can also differentiate into endothelial cells (ECs) or vascular smooth muscle cells (SMCs) under conditioned circumstances [6,7]. Various features, such as scalability and immune privileges, make MSCs ideal candidates for disease therapy. However, their effects are mainly mediated by paracrine factors rather than by cellular differentiation or cell replacement [8]. Among paracrine factors, MSC-derived extracellular vesicles (MSC-EVs) that can mimic the functions of their parental cells have raised significant interest.

EVs are small phospholipid-bilayer-enclosed membrane structures secreted by almost all cell types into the extracellular space [9]. EVs possess the same topology as cells, which present extracellular receptors and ligands, lipids on the outside, and cytoplasmic proteins and RNAs inside. Initially, EVs were thought to be nothing more than the garbage disposal system of cells. Within the past decades, EVs have emerged as important cell–cell communication mediators to regulate multiple physiological and pathological processes [10]. On the one hand, EVs can directly interact with the ligands on recipient cells. On the other hand, EVs carrying a multitude of bioactive molecules and genetic information can be taken up via clathrin-dependent or -independent pathways, such as macropinocytosis, phagocytosis, and caveolae- or lipid-raft-mediated endocytosis, subsequently activating various intracellular signaling cascades in adjacent and remote recipient cells [11]. Based on their size and biogenesis, EVs can be classified into three main subgroups: exosomes, microvesicles, and apoptotic bodies [12,13]. Exosomes are the smallest subtype of EVs; they are intraluminal vesicles formed via invagination of the membrane of multivesicular endosomes (MVEs) and are released upon the fusion of MVEs with the plasma membrane. Microvesicles with diameters of 100–1000 nm are generated upon the direct budding of the plasma membrane, whereas apoptotic bodies are released as a consequence of membrane blebbing during programmed cell death. According to their size alone, EVs can be simply divided into small EVs (sEVs) and medium/large EVs. Due to the difficulty in determining the biogenesis of EVs and the overlap in size among those subsets, the general term “EV” is increasingly used to encompass all different vesicle types in accordance with the Minimum Information for Studies of Extracellular Vesicles 2018 (MISEV2018) guidelines [14]. Indeed, research on EVs has exploded over the past decade, from fundamental biology studies to subjects of significant clinical relevance. Being acellular structures, EVs have extended stability with reduced demands on storage, so they could potentially be lyophilized for off-the-shelf products. As a result, EVs show great potential for being diagnostic and prognostic biomarkers as well as therapeutic agents.

For therapeutic applications, EVs derived from MSCs are mainly adopted [15]. An increasing number of experimental animal and early clinical studies have demonstrated the efficacy of MSC-EV therapy for diseases and disorders of various systems [16,17,18], which has been associated with their inherent vascular-protective and immunomodulatory properties. Moreover, several strategies have been found to be capable of further improving the therapeutic effects and overcoming the disadvantages of MSC-EV-based therapy. Herein, we summarize the current studies on the roles of MSC-EVs in vasculopathies and angiogenesis and discuss strategies to optimize MSC-EV-based therapy for vasculopathies and angiogenesis.

## 2. Therapeutic Potential of Native MSC-EVs for Vasculopathies and Angiogenesis

Accumulating evidence suggests that EVs are strong candidates for cell-free therapy. MSC-EVs not only regulate normal physiological processes, such as angiogenesis, but also have beneficial effects on various pathological processes, such as atherosclerosis (AS), aneurysm, vascular injury, vascular calcification, vascular leakage, pulmonary arterial hypertension (PAH), and diabetic retinopathy (DR). As essential intercellular communication regulators, EVs generated by MSCs can ultimately lead to changes in various functions of vascular ECs, SMCs, and macrophages that reside around the vasculature. Herein, we summarize the therapeutic potential of MSC-EVs for vasculopathies and angiogenesis and outline their underlying molecular mechanisms (Table 1).

### 2.1. Atherosclerosis

AS, characterized by the formation of fibrofatty lesions in the arterial wall, is a chronic inflammatory vascular disease that is the major cause of cardiovascular diseases and stroke. The accumulation of certain plasma lipoproteins, the activation of ECs, and the recruitment of inflammatory cells contribute to the development and progression of AS. MSC-EVs are capable of ameliorating AS by modifying vascular cells to a more normal phenotype and regulating the inflammatory response [19,20,21,22,23]. For example, adipose-derived MSC (ADMSC)-released EV treatment decreased the expression levels of cell adhesion molecules, including vascular cell adhesion protein-1, intercellular adhesion molecule (ICAM)-1, and E-selectin, on the surface of ECs, which were prone to suppress subsequent monocyte adhesion and macrophage accumulation in the vascular walls and finally ameliorated AS in Ldlr^−/−^mice [19]. MSC-EVs can also stabilize atherosclerotic plaques or inhibit lipid deposition in aortic tissues by transferring microRNAs (miRNAs) to SMCs [20]. Macrophage efferocytosis is impaired in atherosclerotic plaques, directly leading to apoptotic cell clearance failure and an alternative inflammation. However, MSC-derived exosomes can restore efferocytosis function and prevent foam cell formation [21]. In addition, MSC-EVs can decrease inflammation by promoting the polarization of M2 macrophages or inhibiting the activation of the NLR family pyrin domain-containing 3 (NLRP3) inflammasome which can lead to macrophage pyroptosis [22,23].

### 2.2. Aneurysm

An aneurysm describes the segmental weakening and dilation of the artery, whose rupture can be life-threatening. Chronic aortic inflammation and inherited and hemodynamic factors play vital roles in the pathogenesis of aneurysmal diseases [54]. MSC-EVs from various sources can mitigate inflammation in experimental aneurysmal disease models. In vivo administration of EVs isolated from umbilical-cord-derived MSCs (UCMSCs) was shown to inhibit aortic inflammation and abdominal aortic aneurysm (AAA) formation in an elastase-treated mouse model, which was dependent on the modulation of immune cells and aortic SMC activation by miR-147 [24]. Exosomes secreted by bone-marrow-derived MSCs (BMMSCs) could maintain the balance of T helper 17/Treg cells to inhibit intracranial aneurysm formation [25]. Moreover, exosomal miR-17-5p from ADMSCs could regulate AAA progression and inflammation by suppressing the thioredoxin-interacting protein–NLRP3 inflammasome [26]. Pathologically, extracellular matrix degradation, which mainly refers to the local destruction of elastin and collagen in the medial layer of the artery, is a hallmark of aneurysms. Notably, MSC-EVs have been demonstrated to attenuate proteolytic activity and impart elastic matrix regenerative benefits in an aneurysmal environment [27].

### 2.3. Vascular Injury

Percutaneous coronary intervention and coronary artery bypass graft surgery are the most widely used therapies for preventing arterial stenosis and occlusion in cardiovascular diseases. However, endothelial dysfunction caused by these invasive surgeries can lead to neointimal hyperplasia or thrombosis, resulting in secondary arterial stenosis and occlusion. MSC-EVs hold promise for endothelial regeneration. For instance, MSC-derived exosomes could enhance the angiogenesis of ECs treated with rapamycin [55]. Additionally, UCMSC-derived exosomes could accelerate re-endothelialization and inhibit vein graft neointimal hyperplasia via the activation of the phosphoinositide 3-kinase/protein kinase B (AKT) and mitogen-activated protein kinase/extracellular signal-regulated kinase-1/2/vascular endothelial growth factor (VEGF) signaling pathways in ECs [28]. In a model of balloon-induced vascular injury, BMMSC-derived exosomes were shown to inhibit the proliferation and migration of vascular SMCs and thus suppress neointimal hyperplasia, which was mediated by exosomal miR-125b that reduce myosin 1E (Myo1e) expression [29].

### 2.4. Vascular Calcification

Vascular calcification is common in the development of AS, hypertension, diabetic vascular disease, vascular injury, and chronic kidney disease. The transition of vascular SMCs from a contractile to an osteoblast-like phenotype under calcifying conditions is a key pathological process in vascular calcification. Increasing evidence has demonstrated that MSC-EVs can alleviate vascular calcification [30,31,32,56]. BMMSC-derived exosomes could alleviate the high-phosphorus-induced calcification of vascular SMCs, as indicated by reduced intracellular calcium content and alkaline phosphatase activity [30]. In a mouse model of chronic kidney disease, treatment with BMMSC-derived exosomes inhibited high-phosphate-induced aortic calcification and ameliorated renal and vascular function via the sirtuin 6 (SIRT6)–high mobility group box 1 deacetylation pathway [31]. In another study, BMMSC-derived exosomes were found to alleviate vascular calcification in chronic kidney disease by delivering enclosed miR-381-3p, which directly targets the nuclear factor of activated T cells 5 [32]. Additionally, the loading of sEVs obtained from human placenta-derived MSCs onto heparin-modified PCL vascular grafts was shown to effectively inhibit the calcification of synthetic vascular grafts and enhance the regeneration of the endothelium and vascular smooth muscle, thus providing long-term vascular patency [56].

### 2.5. Vascular Leakage

As the endothelium serves as an important component of the barrier structure in many organs, impaired endothelial barrier integrity provoked by various physiological and pathophysiological stimuli can cause vascular hyperpermeability and vascular swelling/edema, subsequently resulting in organ injury and dysfunction. Recent studies have reported that MSC-EVs can inhibit endothelial hyperpermeability [33,34,35]. For example, treatment with MSC-derived microvesicles (MSC-MVs) decreased lipopolysaccharide (LPS)-induced endothelial paracellular and transcellular permeabilities by increasing the expression levels of the endothelial intercellular junction proteins VE-cadherin and occludin [33]. In a mouse model of endotoxin-induced acute lung injury (ALI), intratracheal MSC-MV administration restored the pulmonary capillary permeability and attenuated ALI, as attested by decreased Evans blue dye leakage and bronchoalveolar lavage albumin levels [34]. In addition, MSC-EVs could improve the alveolar–capillary barrier integrity via mitochondrial transfer in acute respiratory distress syndrome [35].

### 2.6. Pulmonary Arterial Hypertension (PAH)

PAH is a life-threatening disorder characterized by elevated pulmonary arterial pressure due to increased pulmonary vascular resistance [57]. Vascular pathology, involving endothelial dysfunction, inflammation, and uncontrolled cellular hyperplasia or hypertrophy, may contribute to the luminal narrowing or obstruction of vessels and increased vascular resistance [58]. Multifactorial therapies may be effective for PAH owing to the complex injury processes involved in the pathogenesis of this disorder. Lee et al. reported that MSC-derived exosomes exerted pleiotropic protective effects in a murine model of hypoxic PAH, as demonstrated by the decreased pulmonary influx of macrophages and alleviated lung vascular remodeling [36]. Another study on rats with sugen/hypoxia-induced PAH further demonstrated the roles of MSC-EVs in both preventing and reversing PAH [59]. An improvement in mitochondrial deficits and the activation of the Wnt5a/bone morphogenetic protein signaling pathway are the mechanisms underlying the protective effects of MSC-derived exosomes against PAH [37,38].

### 2.7. Diabetic Retinopathy (DR)

DR is a common complication of diabetes mellitus and a leading cause of vision impairment. Disruptions in normal cell–cell interactions in the retina caused by diabetes mellitus can lead to profound vascular abnormalities, loss of the blood–retinal barrier, and impaired neuronal function [60]. The administration of UCMSC-derived exosomes effectively alleviated the inflammatory response in retinal ECs exposed to high glucose [39]. Another study showed that the treatment of the retinas of DR rats with UCMSC-EVs not only reduced the level of vascular leakage in the retina but also decreased the retinal thickness and associated inflammation [40]. It was reported that UCMSC-EV treatment effectively alleviated retinal oxidative stress and apoptosis of retinal pigment epithelial (RPE) cells to prevent DR progression via the phosphatase and tensin homolog (PTEN)/AKT/nuclear factor erythroid 2-related factor 2 signaling pathway [41]. Moreover, ADMSC-EVs have been reported to shuttle miR-192 to delay the inflammatory response and aberrant angiogenesis in DR by targeting integrin subunit alpha 1 [42]. In addition, BMMSC-derived exosomes were able to ameliorate diabetes-induced retinal injury by suppressing the Wnt/β-catenin signaling pathway and subsequently reducing oxidative stress, inflammation, and angiogenesis [43].

### 2.8. Angiogenesis

Angiogenesis has been an important issue in regenerative medicine. Accumulating evidence suggests that MSC-derived EVs enhance angiogenesis in various organs and tissues. Early or late UCMSC-EV administration restored peripheral pulmonary blood vessel loss and ameliorated pulmonary vascular muscularization in experimental models of bronchopulmonary dysplasia [61]. In addition, a study reported that MSC-derived exosomes promoted the angiogenesis of brain microvascular ECs by increasing the expression of ICAM-1, which in turn promoted the recovery of Parkinson’s disease [62]. In addition to rodents, the delivery of MSC-EVs to non-human primates after myocardial infarction (MI) also enhanced angiogenesis and cardiac recovery without increasing arrhythmia complications, demonstrating the safety and efficacy of MSC-EV-based therapy [63].

MSCs isolated from different sources are not identical. Therefore, the mechanisms determining the pro-angiogenic capability of their EVs may be different. To date, most studies have examined EVs derived from BMMSCs. Activation of VEGF receptors [44] and the nuclear factor-κB pathway [45] in ECs accounts for the pro-angiogenic capability of BMMSC-EVs. Some miRNAs with potential pro-angiogenic roles have been identified in BMMSC-EVs. For example, miR-29b-3p transferred from BMMSC-EVs ameliorated ischemic brain injury via targeted inhibition of PTEN and activation of the AKT signaling pathway, which was involved in the accelerated angiogenesis of brain microvascular ECs and inhibited neuronal apoptosis [46]. Another miRNA, miR-29a, was enriched in BMMSC-EVs and transferred to ECs, promoting angiogenesis in a vasohibin 1-dependent manner [47]. Additionally, proteins contained in BMMSC-EVs have also been verified as important contributors to such modulations. EMMPRIN [48] and Nidogen-1 [49] were demonstrated to be candidate active ingredients for BMMSC-EVs to regulate the migration and angiogenesis potential of ECs, respectively. Regarding ADMSC, Liang et al. reported that ADMSC exosomal miR-125a targeted delta-like 4, resulting in endothelial tip cell formation and angiogenesis [50]. In addition, exosomes isolated from UCMSCs are also able to enhance angiogenesis, in which the activation of hypoxia-inducible factor (HIF)-1α [51] and the activation of the Wnt4/β-catenin pathway [52] are two of the underlying mechanisms. Interestingly, compared to BMMSCs or ADMSCs, it was reported that human endometrium-derived MSCs supported enhanced microvessel density and conferred superior cardioprotection, which was mediated by exosomal miR-21 [53].

Many underlying molecular mechanisms in the therapeutic potential of MSC-EVs for vasculopathies and angiogenesis have been determined, such as functional miRNAs, proteins, mRNAs, and mitochondria in MSC-EVs. However, additional bioactive contents (circular RNAs, long noncoding RNAs, lipids, organelles, etc.) in/on EVs may also participate in their roles. For example, Yan et al. demonstrated that MSC-derived exosomes prevented ischemia-induced pyroptosis in the mouse hindlimb by delivering circHIPK3 [64]. Long noncoding RNA MALAT1 in exosomes was reported to drive regenerative function and modulate inflammation following traumatic brain injury [65]. In addition, increased levels of polyunsaturated fatty acid-containing phospholipids contained within EVs derived from pyruvate kinase muscle isozyme 2-activated T lymphocytes were shown to drive AAA progression by promoting macrophage redox imbalance and migration [66]. Thus, additional bioactive contents contained within MSC-EVs should be investigated to gain comprehensive knowledge about the mechanisms involved in the modulation of vasculopathies and angiogenesis by MSC-EVs.

## 3. Optimization Strategies for MSC-EV-Based Therapy

The efficacy and safety of MSC-EVs in the treatment of vascular diseases and the promotion of angiogenesis have been confirmed by an increasing body of basic research [44,61,62,63,67]. However, native or naked MSC-EVs have limited targeting ability and a low yield. Nowadays, many strategies have been developed to overcome these challenges and further enhance the bioactivity of MSC-EVs, subsequently improving their therapeutic efficacy and efficiency (Figure 1).

### 3.1. Enhancement of MSC-EV Bioactivity

MSC-EVs exert their therapeutic effects mainly through two classes of active ingredients: nucleic acids (especially mRNAs and miRNAs) and proteins (especially surface receptors, intravesicular enzymes, and transcription factors). Therefore, the enrichment of these active ingredients in MSC-EVs via gene modification or cell preconditioning may serve as a promising strategy to enhance their bioactivity.

#### 3.1.1. Gene Modification

Therapeutic agents can be loaded into EVs in two ways: pre-loading (before EV isolation) and post-loading (after EV isolation). Through cell transfection, parental cells can overexpress nucleic acids or proteins and subsequently produce relatively stable EVs loaded with the desired therapeutic agents. Alternatively, cargo can be directly encapsulated into EVs by co-incubation or the temporary permeabilization of the hydrophobic lipid membrane via electroporation, sonication, extrusion, freeze–thaw procedures, or detergent treatment.

Several EV formulations have been established and evaluated for the treatment of vasculopathies and the promotion of angiogenesis. For instance, exosomes obtained from MSCs with the overexpression of Akt [68], stromal-derived factor 1 [69], or HIF-1α [70] are more effective in MI therapy by promoting angiogenesis. Moreover, EVs derived from glial cell line-derived neurotrophic factor-overexpressing MSCs showed a much better ability to ameliorate peritubular capillary loss and tubulointerstitial fibrosis in chronic kidney injury than MSC-EVs, which was associated with the activation of the SIRT1/endothelial nitric oxide synthase (eNOS) signaling pathway [71]. Noncoding RNAs in EVs can also serve as powerful therapeutic agents. Exosomes derived from miR-126-3p-overexpressing MSCs were shown to promote angiogenesis in a diabetic wound model [72] and attenuate vein graft neointimal formation [73]. miR-132-3p promoted the beneficial effects of MSC-derived exosomes in protecting cerebral EC function in ischemic brain injury [74]. Junction adhesion molecule-A (JAM-A, also known as F11R), which is overexpressed in patients with AS, was found to promote atherosclerotic plaque formation by increasing endothelial permeability and allowing mononuclear cell recruitment to the arterial wall [75,76]. Accordingly, miR-145, which targets JAM-A, may have therapeutic potential in the early stage of AS. It was demonstrated that miR-145-overexpressing MSC-EVs efficiently delivered miR-145 to ECs and downregulated JAM-A expression, ultimately reducing atherosclerotic plaque formation [77].

#### 3.1.2. Physical Stimulation

The composition of MSC-EVs is closely associated with the microenvironment in which their original cells reside. Several studies have clearly demonstrated that physical stimulations, such as O_2_ tension, endogenous gasotransmitters, and optical manipulation, could enhance the efficacy of MSC-EVs, which is in consideration of the proteomic and genomic complexities of EVs.

Hypoxic pretreatment is an effective and promising approach for optimizing the therapeutic efficacy of MSC-EVs. It was reported that sEVs obtained from hypoxic MSCs (1% O_2_) had unique characteristics, including enhanced post-ischemic angiogenesis in middle cerebral artery occlusion, which may be mediated by differentially abundant proteins in hypoxic MSC-sEVs that regulate a distinct set of miRNAs linked to angiogenesis [78]. Furthermore, hypoxia preconditioning not only promotes the pro-angiogenic properties of MSC-EVs by transferring specific miRNAs, such as miR-126 or miR-612, but also increases the release of MSC-EVs [79,80]. In addition, exosomes derived from BMMSCs preconditioned with dimethyloxaloylglycine, a small angiogenic molecule that mimics hypoxia in cells by regulating the stability of HIF-1α, can activate the AKT/mechanistic target of rapamycin kinase pathway to exert superior pro-angiogenic activity [81].

Two other endogenous gasotransmitter preconditioning methods also enhance the restorative effects of MSC-EVs. Exosomes released from human placenta-derived MSCs stimulated with nitric oxide (NO) augmented the angiogenic effects of ECs in vitro and ameliorated limb functions in a murine model of hindlimb ischemia, which was mediated by increased VEGF and miR-126 levels in MSC-EVs [82]. The modification of MSC-EVs with hydrogen sulfide (H_2_S) promoted CD45low microglia and CD45high brain mononuclear phagocytes toward a beneficial phenotype, implying their enhanced inflammation modulation potential [83].

As for optical manipulation, exosomes released by UCMSCs exposed to 455 nm blue light are more efficacious in the context of angiogenesis. Mechanistically, blue light illumination of exosomes elevates the levels of miR-135b-5p and miR-499a-3p in ECs [84].

#### 3.1.3. Chemical or Biological Pretreatment

Preconditioning MSCs with chemical factors, like drugs, or biological factors, including components of the cell culture medium, can possibly enhance the biological activities of MSC-EVs. For example, Sung et al. reported that compared to other preconditioning regimens, including H_2_O_2_, LPS, and hypoxia, thrombin preconditioning optimally boosted MSC-EV production and enriched their cargo contents with pro-angiogenic growth factors, such as angiogenin, angiopoietin-1, hepatocyte growth factor, and VEGF [85]. Atorvastatin (ATV) is one of the widely used lipid-lowering drugs for patients with coronary heart disease. ATV pretreatment promoted the therapeutic efficacy of MSC-derived exosomes in acute MI, possibly by promoting EC function. On the one hand, ATV-pretreated MSC-derived exosomes directly accelerated migration and tube-like structure formation and increased the survival of ECs but not cardiomyocytes. On the other hand, ATV-pretreated MSC-derived exosomes protected cardiomyocytes from apoptosis, which was mediated by EC-derived exosomes [86]. In line with this study, ATV-pretreated MSC-derived exosomes also facilitated angiogenesis to accelerate diabetic wound repair, in which the miR-221-3p and AKT/eNOS pathways play important roles [87]. One study reported that exosomes derived from MSCs pretreated with pioglitazone, a common drug used in the treatment of diabetes, promoted the viability and proliferation of ECs injured by high glucose and enhanced angiogenesis in diabetic wound healing [88]. C1q-TNFα-related protein-9 (CTRP9), a novel anti-oxidative cardiokine, is critical in maintaining a healthy local environment in the ischemic heart [89]. Liu et al. reported that CTRP9-281, an active CTRP9 C-terminal polypeptide, was capable of improving cortical bone-derived MSC survival/retention and stimulating the production of VEGFA-rich exosomes from cortical bone-derived MSCs, ultimately exerting superior pro-angiogenic actions [90].

Interestingly, EVs derived from MSCs in inflammatory or infectious environments tend to exhibit anti-inflammatory and immunosuppressive activities. In addition, it is noted that the pro-angiogenic capacity of the EV-encapsulated component of the MSC secretome is not significantly affected by inflammatory stimuli, although the soluble component is strongly inhibited [91]. EVs obtained from MSCs pretreated with TNF-α [92,93], interleukin-1β [94], or LPS [95] can further induce anti-inflammatory M2 macrophage polarization or inhibit the activation of the NLRP3 pathway, implying their enhanced immunomodulatory properties. Therefore, inflammatory-stimuli-pretreated MSC-EVs may exhibit great potential for the treatment of inflammation-related vasculopathies, such as AS and aneurysms.

In addition, it has been reported that the exosomes released from MSCs educated by exosomes from neonatal serum had a better ability to regulate the functions of ECs and enhanced angiogenesis by regulating the AKT/eNOS pathway [96]. Similarly, endothelial differentiation medium preconditioning of ADMSCs upregulated miRNA-31 expression in released microvesicles, thereby promoting angiogenesis [97].

### 3.2. Modulation of MSC-EV Biodistribution

Upon systemic administration in vivo, EVs are generally distributed to organs of the reticuloendothelial system, such as the liver, spleen, and lungs, and undergo rapid clearance [98]. To increase the efficiencyof EVs, several strategies have been developed to increase their stability in the circulation and control their ability to target the lesion area.

#### 3.2.1. Biostability

Considering the short half-life of native EVs in the circulation, it is necessary to develop methods to prevent their rapid clearance by the mononuclear phagocyte system. Modifying EVs with polyethylene glycol (PEG) can enhance their circulation time and improve their opportunity for absorption by targeted cells/tissues [99]. Similarly, MSC-EVs modified with CD47, which is a transmembrane protein that interacts with signal regulatory protein α ligands on the surface of mononuclear macrophages to reduce the phagocytosis of the mononuclear phagocyte system, also exhibit a prolonged retention time in the circulation [100].

#### 3.2.2. Target Delivery

Although integrins and tetraspanins in native EVs are considered to have certain homing and targeting functions, their intrinsic targeting abilities are too weak for practical applications. Many moieties, such as peptides, adaptors, or glycan ligands, can be surface-displayed on EVs to achieve more efficient and selective targeted delivery via specific interactions between the ligands and specific receptors. Both constitutive and pathologically inducible molecules expressed on the endothelial surface can serve as targets for therapeutic delivery. Due to their specific locations in different areas of the vasculature, these molecules can guide the targeted delivery of therapeutics to different organs with vascular disorders. For example, integrin αvβ3, which exhibits a sustained increase in expression on reactive cerebral vascular ECs after ischemia, can be used as a target for therapeutic delivery in ischemic stroke. After intravenous administration to a middle cerebral artery occlusion and reperfusion mouse model, exosomes modified with the cyclo(Arg-Gly-Asp-D-Tyr-Lys) peptide (c[RGDyK]), which exhibits high affinity to integrin αvβ3, significantly accumulated in the lesion region of the ischemic brain [101]. P-selectin-binding peptide (PBP, CDAEWVDVS)-engineered MSC-EVs were shown to ameliorate vascular injury and protect against ischemic acute kidney injury in the early stage by selectively binding to injured renal ECs [102]. Moreover, the CAR (CARSKNKDC) peptide specifically targeting hypertensive pulmonary arteries represents a feasible choice for efficient delivery in PAH therapy [103]. Apart from proteins and peptides, the glycocalyx that constitutes the outermost layer of EVs can also be engineered to endow EVs with an active targeting ability. During inflammation, E-selectin is expressed on activated ECs and mediates leukocyte adhesion [104]. Sialyl Lewis-X (sLeX) is a tetrasaccharide glycan ligand of E-selectin, and the surface display of sLeX on EVs is a viable strategy for tailoring EV specificity for activated ECs. When this reconfigurable glycoengineering strategy was applied to MSC-EVs, the EC-targeting ability of sLeX was combined with the intrinsic anti-inflammatory activity of MSC-EVs, achieving higher efficacy in attenuating endothelial damage [105]. Furthermore, aptamer-guided EV theranostics has been developed for oncology [106]; however, these active targeting units with low molecular weights need to be explored for vascular applications. In addition, EVs can also be modified by loading iron oxide nanoparticles into their cavities to permit targeting to desired locations under magnetic manipulation [107].

#### 3.2.3. Controlled Release

It is acknowledged that the delivery route and dosage influence the biodistribution pattern of EVs [98]. Local injection of EVs directly into injured sites helps to decrease unanticipated side effects that may be caused by systemic administration. A number of routes, including intramyocardial, intratracheal, renal capsule, and intravitreal injections, have been demonstrated to be effective and safe for EV delivery. Moreover, biomaterial-based delivery systems can be applied to further improve the therapeutic efficiency and effects of local EV administration by retaining EVs at the injured sites. For example, Zhang et al. reported that chitosan hydrogel not only notably augmented the retention of exosomes in ischemic hindlimbs but also increased the stability of exosomal proteins and miRNAs that contribute to the functionality of MSC-derived exosomes [108].

Among biomaterials, injectable and heat-sensitive hydrogels based on different compositions, such as hyaluronic acid, alginate, chitosan, collagen, and amphiphilic peptides, are the most widely used for EV delivery. Several studies have demonstrated enhanced angiogenic and immunosuppressive effects of MSC-EVs incorporated into biocompatible hydrogels in the context of hindlimb ischemia [108], chronic wounds [109], MI [110], and fracture [51]. For certain administrations, an appropriate hydrogel composition may be selected after considering its biomedical history, pore size, degradation, in situ jellification profile, and mechanical and release properties. For instance, an antibacterial polypeptide-based FHE hydrogel (F127/OHA-EPL) is suitable for delivering MSC-derived exosomes to promote wound healing [109], while self-assembling peptide hydrogels composed of cardiac-protective peptides are suitable for encapsulating MSC-derived exosomes to treat MI [110]. MSC-EVs can also be added to implantable scaffolds to exert a synergistic effect on vascular disorders. The incorporation of MSC-EVs exerted a significant synergistic effect in increasing the patency of vascular grafts [111]. In another study, Ko et al. demonstrated efficient kidney tissue regeneration using an integrated bioactive scaffold system functionalized with pro-regenerative MSC-EVs and other biochemical and biophysical cues [112]. Additionally, 3D printing technology provides a prospective method for fabricating scaffolds with desired structures using bioinks composed of MSC-derived exosomes and different hydrogels [113].

Controlling the release rate and order of MSC-EVs from hydrogels is a promising strategy for achieving optimal therapeutic outcomes. EVs are released from hydrogels mainly through diffusion. Therefore, the release rate of EVs is mainly affected by the molecular size of the EVs and the pore size of the hydrogels. Recently, dynamic cross-links have been introduced into polymer networks to fabricate hydrogels displaying the controlled and on-demand release of EVs in specific injured tissues, imparting hydrogels with the ability to sense and respond to the surrounding microenvironment. Enzyme-degradable hydrogels are among the most extensively studied dynamic hydrogels. Several enzyme-degradable hydrogels have potential applications in vascular therapeutics. For example, an injectable matrix metalloproteinase-2 (MMP2)-sensitive self-assembling peptide (KMP2) hydrogel has been demonstrated to mediate the sustained release of MSC-EVs for promoting angiogenesis and improving the overall renal function in renal ischemia–reperfusion injury [114]. Since the activity of MMP2 was elevated in ischemic renal tissues, the cross-linked KMP2 hydrogels degraded with increased pore sizes after injection into the renal capsule, which promoted the greater release of MSC-EVs with preserved structure and bioactivity and ultimately led to enhanced therapeutic efficacy. To prevent in-stent restenosis, Zou et al. developed a vascular stent capable of releasing MSC-derived exosomes in response to vascular inflammatory marker lipoprotein-associated phospholipase A2 to resolve chronic inflammation, repress the overproliferation of SMCs, and reduce the risk of delayed re-endothelialization [21]. In addition, MSC-derived exosome-eluting stents responsive to reactive oxygen species (ROS) also exhibit therapeutic potential for a wide range of vascular diseases [115]. Both local inflammation after AS and mechanical injury from stent deployment induce elevated levels of ROS; therefore, ROS-triggered vascular stents can not only regulate vascular remodeling and inflammation but also promote angiogenesis in ischemic diseases. Bilayered thiolated alginate/PEG diacrylate hydrogels loaded with EVs have been fabricated to promote rapid and scarless wound healing. Different thiolated alginate/PEG diacrylate (SASH/PEG-DA) ratios endowed the hydrogels with different mechanical strengths and degradation rates. EVs secreted by BMMSCs that could promote angiogenesis and collagen deposition, as well as EVs derived from miR-29b-3p-enriched BMMSCs that were able to suppress capillary proliferation and collagen deposition, were sequentially released from the lower and upper layers of the bilayered hydrogel, aiming at different wound repair phases to achieve rapid and scarless wound healing [116].

### 3.3. Improvement in MSC-EV Yield

Preclinical and clinical research on EV-based therapy requires large quantities of MSC-EVs, which cannot be easily satisfied in routine culture conditions. Therefore, new methods are required to address the poor yield and scalability issues of MSC-EVs without sacrificing their functionality.

The cell source, cell passage, cell seeding density, medium composition, EV collection frequency, and even substrate topography are known to impact the production and bioactivity of EVs [117,118]. Recently, plenty of studies have demonstrated that specific physical stimulations, including mechanical force [119] or exposure to hypoxia [91,120], certain chemicals such as thrombin [85], adiponectin [121], metformin [122], or small molecule modulators [123], and treatment with some biomaterials like bioglasses [124] or nanoparticles [125,126], effectively boost the output of MSC-EVs without altering their intrinsic therapeutic potential. Additionally, spheroid and three-dimensional (3D) dynamic cultures have been found to significantly promote EV secretion [127,128,129]. Appropriate combinations of these methods possess a stronger capability to enhance the production of EVs. Based on the processes and regulatory mechanisms of the biogenesis and release of EVs, several novel strategies have been proposed and demonstrated to stimulate EV production. For example, by screening genes related to exosome biogenesis and secretion in cells, Kojima et al. developed an exosome production booster that consists of STEAP3, syndecan-4, and a fragment of l-aspartate oxidase [130]. These genetically encoded devices are capable of increasing the exosome yield without changing their size distribution and are also confirmed to be effective in human MSCs. In addition, an intracellular increase in Ca^2+^ induces EV secretion through the outward budding of the plasma membrane. Using extended mitochondrial targeting domain (eMTDΔ4) peptides that can trigger intracellular Ca^2+^ influx, a novel method was developed to obtain increased production of high-purity EVs from UCMSCs [131].

Apart from directly enhancing the EV production capability of cells, achieving large-scale manufacturing and improving isolation methods of EVs are other strategies for obtaining a sizable number of EVs. Compared to the traditional cell culture dish, flask, or cell factory, a bioreactor is a preferable type of cell culture vessel for the large-scale expansion of MSCs and large-volume harvesting of cell culture media [132]. On the one hand, the use of bioreactors to generate EVs requires less labor and has a low risk of contamination. On the other hand, the aforementioned preconditioning methods to increase EV production or bioactivity can be perfectly applied in both microcarrier and hollow-fiber bioreactors. For instance, the oxygen supply or chemical concentration can be easily controlled and monitored in bioreactors. Moreover, shear stress can be induced using a spinner in bioreactors. In addition, different biomaterials can be made into microcarriers to affect the cell growth kinetics and phenotype, subsequently influencing EV secretion and possibly their therapeutic potential. In addition to upstream processing for producing EV-containing conditioned media, the optimization of EV isolation in downstream processing should not be ignored. In recent years, various methods based on the density, size, surface charge, and molecular composition have been proposed for EV purification and concentration, which comprise differential ultracentrifugation, ultrafiltration, tangential flow filtration (TFF), PEG precipitation, anion-exchange chromatography (AEX), affinity chromatography, and size-exclusion chromatography (SEC) [133]. In particular, TFF and AEX, which are capable of processing large volumes of conditioned media, are considered promising methods for scalable EV isolation.

## 4. Conclusions and Perspectives

MSC-EVs appear to have a promising future in providing cell-free therapies for a variety of diseases, which is associated with their protective effects on the vasculature. MSC-EVs exhibit therapeutic potential for various vasculopathies by improving the functions of several vascular cells and components residing around the blood vessels. To date, substantial advances have been made in the establishment of strategies for further improving the therapeutic efficacy and efficiency of MSC-EVs. However, several challenges still need to be addressed to facilitate their clinical translation.

Most medical therapies are highly consistent in terms of their composition and bioactivity; however, EV formulations are highly variable. As EVs are important constituents of the cellular secretome, both the cell source and culture conditions considerably impact the components of EVs. Moreover, different isolation methods lead to differences in the purity, characteristics, compositions, and even biological functions of EV products [134]. Highly variable EV components, which are difficult to comprehensively characterize, greatly hinder EVs from providing standardized, repeatable, and reliable therapy. Therefore, it is critical to define and standardize the methods for EV separation, purification, characterization, and storage to improve homogeneity among EV preparations, which can facilitate inter-laboratory validation and reproducibility studies, as well as achieve good manufacturing practices. The relevant guidelines proposed by the scientific community are an important step in this direction [8,14]. As each isolation technique has its own strengths and limitations, further technological progress is needed for large-scale clinical-grade EV production.

In fact, EVs seem like complex cocktails of therapeutic agents. In order to achieve more efficacious and safe precision-engineered MSC-EV-based therapeutics, several issues should be taken into consideration. First of all, the optimal MSC source and conditions for EV production must be defined. As MSCs derived from different sources may release EVs with distinct contents, it is important to ensure which ones provide superior therapeutic effects for certain vascular disorders. The definition of donor inclusion/exclusion criteria is also required to prevent obtaining MSC-EVs with impaired therapeutic effects due to the existing comorbidities and/or age of the donor. Then, it is necessary to clarify the mechanisms of EV biogenesis and uptake, as well as identify the functional cargoes and activated intracellular mechanisms that contribute to MSC-EV-mediated vascular repair in a comprehensive manner. Moreover, a better understanding of what gene modification and preconditioning strategies bring to MSCs and how they promote the bioactivity of MSC-EVs will promote the development of more effective and consistent EV preparations with fewer side effects. Last but not least, further progress is needed to determine the best time window, route, dosage, and frequency of EV administration. After administration, the pharmacokinetics and long-term effects of MSC-EVs in the body should be determined. The current development of the human kidney and liver organoid-based multi-organ-on-a-chip model [135], as well as various methods for EV labeling and tracking, such as fluorescent dye labeling [136], radioisotope labeling [137], and reporter systems [138], suggests a promising future in this area.

## Figures and Tables

**Figure 1 biomolecules-13-01109-f001:**
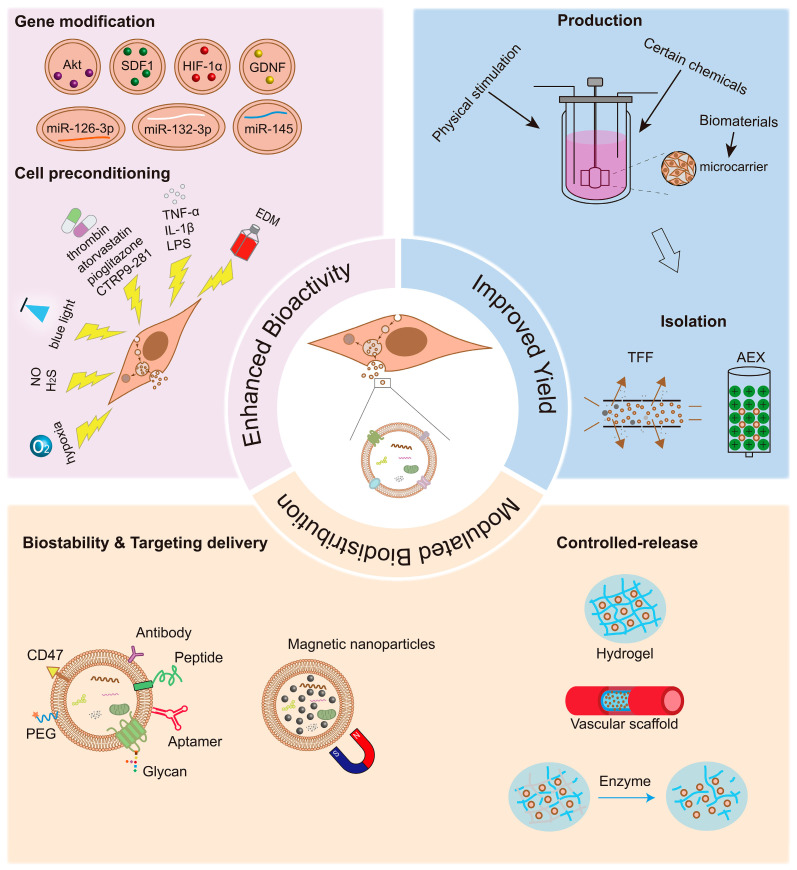
Optimization strategies for mesenchymal stromal cell-derived extracellular vesicle (MSC-EV)-based therapy in vasculopathies and angiogenesis. Genetic modifications, physical, chemical, and biological strategies, and biomaterial stimulation can be used to improve the therapeutic effect of MSC-EVs. The enhanced therapeutic effects may depend on the superior bioactivity, high yield, efficient delivery, and controlled release of MSC-EVs to the desired regions. Akt: protein kinase B; SDF1: stromal-derived factor 1; HIF-1α: hypoxia-inducible factor-1α; GDNF: glial cell line-derived neurotrophic factor; TNF-α: tumor necrosis factor-α; IL-1β: interleukin-1β; LPS: lipopolysaccharide; CTRP9: C1q-TNFα related protein-9; EDM: endothelial differential medium; TFF: tangential flow filtration; AEX: anion-exchange chromatography; PEG: polyethylene glycol.

**Table 1 biomolecules-13-01109-t001:** Mechanisms involved in the modulation of vasculopathies and angiogenesis by mesenchymal stromal cell-derived extracellular vesicles.

Application	EV Source	Target Cells	EV Cargo	Proposed Mechanisms	Ref.
Atherosclerosis	mADMSC	EC	-	MAPK and NF-κB pathways	[19]
	hBMMSC	SMC	miR-221	NAT1	[20]
	rBMMSC	Macrophage	-	SLC2a1, STAT3/RAC1, and CD300a pathways; CD36-mediated pathway	[21]
	mBMMSC	Macrophage	miR-let7	HMGA2/NF-kB pathway; IGF2BP1/PTEN pathway	[22]
	mBMMSC	Macrophage	miR-223	NLRP3	[23]
Aneurysm	UCMSC	Macrophage, SMC	miR-147	HMGB-1 and IL-17	[24]
	hBMMSC	-	miR- 23b-3p	PI3k/Akt/NF-κB pathway, KLF5	[25]
	mADMSC	Macrophage	miR-17-5p	TXNIP	[26]
	hBMMSC	SMC	-	MMP, TIMP	[27]
Vascular injury	UCMSC	EC	-	PI3K/AKT and MAPK/ERK1/2 pathways, VEGF	[28]
	rBMMSC	SMC	miR-125b	Myo1e	[29]
Vascular calcification	hBMMSC	SMC	-	NONHSAT 084969.2/NF-κB pathway	[30]
	mBMMSC	SMC	SIRT6	HMGB1	[31]
	BMMSC	SMC	miR-381-3p	NFAT5	[32]
Vascular leakage	mBMMSC	EC	HGF	VE-cadherin and occludin	[33]
	hBMMSC	EC, macrophage	Angiopoietin-1	-	[34]
	hBMMSC	EC	mitochondria	-	[35]
Pulmonary arterial hypertension	hBMMSC, UCMSC	-	-	STAT3 pathway	[36]
	hBMMSC	SMC	-	SIRT4	[37]
	UCMSC	EC, SMC	-	Wnt5a/BMP pathway	[38]
Diabetic retinopathy	UCMSC	EC	miR-126	HMGB1 pathway	[39]
	UCMSC	EC	miR-18	MAP3K1/NF-κB pathway	[40]
	UCMSC	RPEC	NEDD4	PTEN, pAKT, NRF2	[41]
	rADMSC	RPEC, Müller cell, EC	miR-192	ITGA1	[42]
	rBMMSC	-	-	Wnt/β-catenin pathway	[43]
Angiogenesis	mBMMSC	EC	-	VEGFR1, VEGFR2	[44]
	hBMMSC	EC	-	NF-κB pathway	[45]
	rBMMSC	EC	miR-29b-3p	PTEN, Akt	[46]
	mBMMSC	EC	miR-29a	VASH1	[47]
	hBMMSC	EC	EMMPRIN	-	[48]
	rBMMSC	EC	Nidogen1	Myosin-10	[49]
	hADMSC	EC	miR-125a	DDL4	[50]
	UCMSC	EC	-	HIF-1α	[51]
	UCMSC	EC	Wnt4	β-Catenin	[52]
	hENMSC	EC	miR-21	-	[53]

EV: extracellular vesicle; mADMSC: mouse adipose-derived mesenchymal stromal cell; hBMMSC: human bone marrow-derived mesenchymal stromal cell; rBMMSC: rat bone marrow-derived mesenchymal stromal cell; mBMMSC: mouse bone marrow-derived mesenchymal stromal cell; UCMSC: human umbilical cord-derived mesenchymal stromal cell; rADMSC: rat adipose-derived mesenchymal stromal cell; hADMSC: human adipose-derived mesenchymal stromal cell; hENMSC: human endometrium-derived mesenchymal stromal cell; RPEC: retinal pigment epithelial cell.

## Data Availability

Not applicable.

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
