# Peer review of "Mesenchymal Stromal Cell-Derived Extracellular Vesicles for Vasculopathies and Angiogenesis: Therapeutic Applications and Optimization"

_biomolecules, 2023, doi:10.3390/biom13071109_

Round 1
Reviewer 1 Report
This is an interesting and well written review of the extracellular vesicles of MSC origin, their uses, design, and production. However, the serious limitations of the EV approach received limited discussion. Reproducibility of EV components and therapeutic results is a major concern, for instance, given the highly complex components and their variability depending upon the cell source and production techniques. Has any study compared the therapeutic effectiveness of EVs derived from varied sources and production techniques? How do the vesicle components vary when produced with the same cell source under different culture conditions and passage number? The mechanism of therapeutic action in the reviewed studies is also a major concern. Table 1 lists many proposed mechanisms for many studies. But this must be discussed in light of the highly complex mixture of components contained within the EVs.
The English language is good but needs some general improvement.
Author Response
Point 1: This is an interesting and well written review of the extracellular vesicles of MSC origin, their uses, design, and production. However, the serious limitations of the EV approach received limited discussion. Reproducibility of EV components and therapeutic results is a major concern, for instance, given the highly complex components and their variability depending upon the cell source and production techniques. Has any study compared the therapeutic effectiveness of EVs derived from varied sources and production techniques?
Response 1: Thanks for the thoughtful comments. We agree with the opinion that reproducibility of EV components and therapeutic results is a major concern. As a result, we have emphasized the importance to define and standardize the methods for EV separation, purification, characterization, and storage to improve homogeneity within EV preparations in the section “4. Conclusion and perspectives” (lines 553 to 556 of page 13). We have also suggested that it is important to ensure MSC-EVs from which source provide superior therapeutic effects for certain vascular disorders in the section “4. Conclusion and perspectives” (lines 562 to 567 of page 14).
There are many studies compared the therapeutic effectiveness of EVs derived from varied sources and production techniques, some of which are listed below. For example, the osteoinductivity of MSC-derived exosomes produced from different sources (bone mesenchymal stem cells and adipose-derived stem cells) and conditions were compared for an optimized osteogenic MSC-derived exosome in a study (Liu A et al., Biomaterials, 2021). As for production techniques, the physicochemical characteristics, yield, and activity of exosomes derived from two-dimensional (2D) or 3D cultures, as well as produced by differential ultracentrifugation or tangential flow filtration, were compared (Haraszti et al., Mol Ther, 2018). Additionally, wound healing and anti-inflammatory potential of MSC-EVs generated with the natural EV method or a new method using extended mitochondrial targeting domain peptide were compared (Lim et al., Journal of extracellular vesicles, 2022 ).
References
Liu, A.; Lin, D.; Zhao, H.; Chen, L.; Cai, B.; Lin, K.; Shen, S.G. Optimized BMSC-derived osteoinductive exosomes immobilized in hierarchical scaffold via lyophilization for bone repair through Bmpr2/Acvr2b competitive receptor-activated Smad pathway. Biomaterials 2021, 272, 120718, doi:10.1016/j.biomaterials.2021.120718.
Haraszti, R.A.; Miller, R.; Stoppato, M.; Sere, Y.Y.; Coles, A.; Didiot, M.C.; Wollacott, R.; Sapp, E.; Dubuke, M.L.; Li, X.; et al. Exosomes Produced from 3D Cultures of MSCs by Tangential Flow Filtration Show Higher Yield and Improved Activity. Mol Ther 2018, 26, 2838-2847, doi:10.1016/j.ymthe.2018.09.015.
Lim, K.M.; Han, J.H.; Lee, Y.; Park, J.; Dayem, A.A.; Myung, S.H.; An, J.; Song, K.; Kang, G.H.; Kim, S.; et al. Rapid production method with increased yield of high-purity extracellular vesicles obtained using extended mitochondrial targeting domain peptide. Journal of extracellular vesicles 2022, 11, e12274, doi:10.1002/jev2.12274.
Point 2: How do the vesicle components vary when produced with the same cell source under different culture conditions and passage number? The mechanism of therapeutic action in the reviewed studies is also a major concern. Table 1 lists many proposed mechanisms for many studies. But this must be discussed in light of the highly complex mixture of components contained within the EVs.
Response 2: Thank the reviewer for the professional suggestion. In the sections “3.1.2. Physical stimulation” and “3.1.3. Chemical or biological pretreatment” (page 9 to 10), we have reviewed the important nucleic acid or protein components in vesicles derived from the same cell source under different culture conditions, which account for the enhanced bioactivity of these MSC-EVs. Cell passage number also impacts the therapeutic effect and vesicle components of EVs. Patel et al. only reported that cell passage number impacts MSC-EV vascularization bioactivity without clarifing the involved mechanisms (Patel et al., Bioeng Transl Med, 2017). Another study identified upregulated miR-122-5p and miR-146a-5p within sEVs of replicative senescent umbilical cord mesenchymal stem cells (Kim et al., Bioengineering & translational medicine, 2023). It has also been reported that miR-21 and miR-217 enriched in EVs derived from replicative senescent HUVECs contribute to the prosenescence microenvironment (Mensà et al., Journal of extracellular vesicles, 2020).
Actually, the components of EVs contain highly complex mixture of bioactive molecules and even organelles. In addition to the reviewed miRNAs, proteins, mRNAs, and mitochondria in our manuscript, other bioactive contents (circRNAs,lncRNAs, lipids, organelles, etc.) in/on MSC-EVs may also participate in their therapeutic effects for vasculopathies and angiogenesis. Thus, we have added a paragraph for supplement at the end of the section “2. Therapeutic potential of native MSC-EVs for vasculopathies and angiogenesis”. “Many underlying molecular mechanisms in the therapeutic potential of MSC-EVs for vasculopathies and angiogenesis have been determined, such as the functional miRNAs, proteins, mRNAs, and mitochondria in MSC-EVs. However, additional bioactive contents (circular RNAs, long noncoding RNAs, lipids, organelles, etc.) in/on EVs may also participate in their roles. For example, Yan et al. demonstrated that MSC-derived exosomes prevented ischemia-induced pyroptosis in mouse hindlimb by delivering circHIPK3 [64]. Long noncoding RNA MALAT1 in exosomes was reported to drive regenerative function and modulate inflammation following traumatic brain injury [65]. In addition, increased levels of polyunsaturated fatty acid-containing phospholipids contained within EVs derived from pyruvate kinase muscle isozyme 2-activated T lymphocytes was shown to drive AAA progression by promoting macrophage redox imbalance and migration [66]. Thus, additional bioactive contents contained within the MSC-EVs should be investigated for having a comprehensive knowledge about the mechanisms involved in the modulation of vasculopathies and angiogenesis by MSC-EVs.” (lines 256 to 269 of page 7).
References
Patel, D.B.; Gray, K.M.; Santharam, Y.; Lamichhane, T.N.; Stroka, K.M.; Jay, S.M. Impact of cell culture parameters on production and vascularization bioactivity of mesenchymal stem cell-derived extracellular vesicles. Bioengineering & translational medicine 2017, 2, 170-179, doi:10.1002/btm2.10065.
Kim, C.G.; Lee, J.K.; Cho, G.J.; Shin, O.S.; Gim, J.A. Small RNA sequencing of small extracellular vesicles secreted by umbilical cord mesenchymal stem cells following replicative senescence. Genes & genomics 2023, 45, 347-358, doi:10.1007/s13258-022-01297-y.
Mensà, E.; Guescini, M.; Giuliani, A.; Bacalini, M.G.; Ramini, D.; Corleone, G.; Ferracin, M.; Fulgenzi, G.; Graciotti, L.; Prattichizzo, F.; et al. Small extracellular vesicles deliver miR-21 and miR-217 as pro-senescence effectors to endothelial cells. Journal of extracellular vesicles 2020, 9, 1725285, doi:10.1080/20013078.2020.1725285.
- Yan, B.; Zhang, Y.; Liang, C.; Liu, B.; Ding, F.; Wang, Y.; Zhu, B.; Zhao, R.; Yu, X.Y.; Li, Y. Stem cell-derived exosomes prevent pyroptosis and repair ischemic muscle injury through a novel exosome/circHIPK3/ FOXO3a pathway. Theranostics 2020, 10, 6728-6742, doi:10.7150/thno.42259.
- Patel, N.A.; Moss, L.D.; Lee, J.Y.; Tajiri, N.; Acosta, S.; Hudson, C.; Parag, S.; Cooper, D.R.; Borlongan, C.V.; Bickford, P.C. Long noncoding RNA MALAT1 in exosomes drives regenerative function and modulates inflammation-linked networks following traumatic brain injury. Journal of neuroinflammation 2018, 15, 204, doi:10.1186/s12974-018-1240-3.
Dang, G.; Li, T.; Yang, D.; Yang, G.; Du, X.; Yang, J.; Miao, Y.; Han, L.; Ma, X.; Song, Y.; et al. T lymphocyte-derived extracellular vesicles aggravate abdominal aortic aneurysm by promoting macrophage lipid peroxidation and migration via pyruvate kinase muscle isozyme 2. Redox biology 2022, 50, 102257, doi:10.1016/j.redox.2022.102257.
Reviewer 2 Report
In the manuscript (Manuscript ID: biomolecules-2467875) entitled “Mesenchymal Stromal Cell-derived Extracellular Vesicles for Vasculopathies and Angiogenesis: Therapeutic Applications and Optimization” by Ying Zhu, Zhao-fu Liao, Miao-hua Mo, Xing-dong Xiong, the authors review the role of mesenchymal stem cell-derived extracellular vesicles, and their therapeutic potential in angiogenesis and different vasculopathies, including atherosclerosis, aneurysm, vascular injury, vascular calcification, vascular leakage, pulmonary arterial hypertension and diabetic retinopathy. Next, they consider strategies for optimizing their therapeutic effects, such as gene modification, physical stimulation, chemical or biological pretreatment, modulation of distribution, biostability and target delivery, controlled release, and improvement of their yield. This work is well-written and structured. I only have two comments.
a) The Abstract should include the strategies considered for the optimization of therapeutic effects of mesenchymal stromal cell-derived extracellular vesicles.
b) Although the authors refer that the minimal defining criteria of the International Society for Cellular Therapy, MSCs must be plastic adherent, express CD105, CD73, and CD90 but not CD45, CD34, CD14, or CD11b, CD79α or CD19, resident (native) MSCs can express markers that change during culture passages. Thus, it is important to make it clear in this section that the resident (in situ) mesenchymal stromal cells are CD34+, considered CD34+ stromal cells/fibroblasts/fibrocytes/telocytes, which lose CD34 expression in successive culture passages. In addition, resident/native CD34+ stromal cells/progenitor cells lose expression of CD34 and are a source of alpha+ SMA myofibroblasts during different stages of repair. In these stages of angiogenesis and repair, extracellular multivesicular bodies play an important role.
Author Response
Point 1: In the manuscript (Manuscript ID: biomolecules-2467875) entitled “Mesenchymal Stromal Cell-derived Extracellular Vesicles for Vasculopathies and Angiogenesis: Therapeutic Applications and Optimization” by Ying Zhu, Zhao-fu Liao, Miao-hua Mo, Xing-dong Xiong, the authors review the role of mesenchymal stem cell-derived extracellular vesicles, and their therapeutic potential in angiogenesis and different vasculopathies, including atherosclerosis, aneurysm, vascular injury, vascular calcification, vascular leakage, pulmonary arterial hypertension and diabetic retinopathy. Next, they consider strategies for optimizing their therapeutic effects, such as gene modification, physical stimulation, chemical or biological pretreatment, modulation of distribution, biostability and target delivery, controlled release, and improvement of their yield. This work is well-written and structured. I only have two comments.
- a) The Abstract should include the strategies considered for the optimization of therapeutic effects of mesenchymal stromal cell-derived extracellular vesicles.
Response 1: Thank the reviewer very much for the comment. As suggested by the reviewer, we have revised “We also discuss the current strategies to optimize MSC-EV-based therapy for vasculopathies and angiogenesis and the challenges that need to be overcome to allow their broad clinical translation.” as “We also discuss the current strategies to optimize their therapeutic effects, that depend on the superior bioactivity, high yield, efficient delivery, and controlled release of MSC-EVs to the desired regions, as well as the challenges that need to be overcome to allow their broad clinical translation.” (lines 22 to 25 of page 1).
Point 2: b) Although the authors refer that the minimal defining criteria of the International Society for Cellular Therapy, MSCs must be plastic adherent, express CD105, CD73, and CD90 but not CD45, CD34, CD14, or CD11b, CD79α or CD19, resident (native) MSCs can express markers that change during culture passages. Thus, it is important to make it clear in this section that the resident (in situ) mesenchymal stromal cells are CD34+, considered CD34+ stromal cells/fibroblasts/fibrocytes/telocytes, which lose CD34 expression in successive culture passages. In addition, resident/native CD34+ stromal cells/progenitor cells lose expression of CD34 and are a source of alpha+ SMA myofibroblasts during different stages of repair. In these stages of angiogenesis and repair, extracellular multivesicular bodies play an important role.
Response 2: Thank the reviewer for reminding us of one important fact that CD34 being expressed in tissue-resident MSC, and its negative finding being a consequence of cell culturing. We have revised “According to the minimal defining criteria of the International Society for Cellular Therapy, MSCs must be plastic adherent, express CD105, CD73 and CD90 but not CD45, CD34, CD14 or CD11b, CD79α or CD19 and human leukocyte antigen-DR surface molecules, and possess tri-lineage differentiation potential [2].” as “According to the minimal defining criteria of the International Society for Cellular Therapy, cultured MSCs must be plastic adherent, express CD105, CD73 and CD90 but not CD45, CD34, CD14 or CD11b, CD79α or CD19 and human leukocyte antigen-DR surface molecules, and possess tri-lineage differentiation potential [2]. In addition, tissue-resident MSCs are considered CD34+ stromal cells/fibroblasts/fibrocytes/telocytes and lose CD34 expression in successive culture passages [3,4]. Resident/native CD34+ stromal cells/progenitor cells lose expression of CD34 and are a source of alpha+ SMA myofibroblasts during different stages of repair [5].” (lines 38 to 45 of page 1 to 2).
References
- Díaz-Flores, L.; Gutiérrez, R.; García, M.P.; Sáez, F.J.; Díaz-Flores, L., Jr.; Valladares, F.; Madrid, J.F. CD34+ stromal cells/fibroblasts/fibrocytes/telocytes as a tissue reserve and a principal source of mesenchymal cells. Location, morphology, function and role in pathology. Histology and histopathology 2014, 29, 831-870, doi:10.14670/hh-29.831.
- Lin, C.S.; Ning, H.; Lin, G.; Lue, T.F. Is CD34 truly a negative marker for mesenchymal stromal cells? Cytotherapy 2012, 14, 1159-1163, doi:10.3109/14653249.2012.729817.
- Díaz-Flores, L.; Gutiérrez, R.; García, M.P.; González, M.; Sáez, F.J.; Aparicio, F.; Díaz-Flores, L., Jr.; Madrid, J.F. Human resident CD34+ stromal cells/telocytes have progenitor capacity and are a source of αSMA+ cells during repair. Histology and histopathology 2015, 30, 615-627, doi:10.14670/hh-30.615.
Round 2
Reviewer 1 Report
The conclusion section needs to include a statement akin to the following: "Most medical therapies are highly consistent in terms of their composition and bioactivity. In contrast, the highly complex EV components vary considerably by cell source, culture conditions and process of harvesting. The multi-component EVs are also difficult to comprehensively characterize due to their high complexity and variability. The high sensitivity of EVs to their experimental conditions represents a major challenge for producing a standardized, repeatable, and reliable therapy."
The English language quality is ok. Will require minor editing.
Author Response
Response to Reviewer 1 Comments
Point 1: The conclusion section needs to include a statement akin to the following: "Most medical therapies are highly consistent in terms of their composition and bioactivity. In contrast, the highly complex EV components vary considerably by cell source, culture conditions and process of harvesting. The multi-component EVs are also difficult to comprehensively characterize due to their high complexity and variability. The high sensitivity of EVs to their experimental conditions represents a major challenge for producing a standardized, repeatable, and reliable therapy."
Response 1: Thank the reviewer very much for the suggestion. As suggested by the reviewer, we have revised “Methods for obtaining EVs vary among studies, however, different methods lead to differences in purity, characteristics, compositions and even biological functions of MSC-EVs [134]. Therefore, it is critical to define and standardize the methods for EV separation, purification, characterization, and storage to improve homogeneity within EV preparations, which can facilitate inter-laboratory validation and reproducibility studies as well as achieve good manufacturing practices.” as “Most medical therapies are highly consistent in terms of their composition and bioactivity, however, EV formulations are highly variable. As EV is an important composition of cellular secretome, both cell source and culture condition considerably impact the components of EVs. Moreover, different obtaining methods lead to differences in purity, characteristics, compositions and even biological functions of EV products [134]. The highly variable EV components, that are difficult to comprehensively characterize, greatly hinder EVs from providing a standardized, repeatable, and reliable therapy. Therefore, it is critical to define and standardize the methods for EV separation, purification, characterization, and storage to improve homogeneity within EV preparations, which can facilitate inter-laboratory validation and reproducibility studies as well as achieve good manufacturing practices.” (lines 551 to 561 of page 13 to 14).
Round 3
Reviewer 1 Report
The authors have been very responsive to my concerns and have revised the manuscript accordingly. I have no further concerns.
The English language quality is acceptable. There are only minor edits needed.